# Investigation of Direct Electron Transfer of Glucose Oxidase on a Graphene-CNT Composite Surface: A Molecular Dynamics Study Based on Electrochemical Experiments

**DOI:** 10.3390/nano14131073

**Published:** 2024-06-24

**Authors:** Taeyoung Yoon, Wooboum Park, Juneseok You, Sungsoo Na

**Affiliations:** 1Department of Mechanical Engineering, Korea University, Seoul 02841, Republic of Korea; tyler1111@korea.ac.kr (T.Y.); pwb2002@korea.ac.kr (W.P.); 2Department of Mechanical Engineering, Kumoh National Institute of Technology, Gumi 39177, Republic of Korea

**Keywords:** direct electron transfer, glucose oxidase, graphene, carbon nanotube electrochemistry, enzyme coating, molecular dynamics

## Abstract

Graphene and its variants exhibit excellent electrical properties for the construction of enzymatic interfaces. In particular, the direct electron transfer of glucose oxidase on the electrode surface is a very important issue in the development of enzyme-based bioelectrodes. However, the number of studies conducted to assess how pristine graphene forms different interfaces with other carbon materials is insufficient. Enzyme-based electrodes (formed using carbon materials) have been extensively applied because of their low manufacturing costs and easy production techniques. In this study, the characteristics of a single-walled carbon nanotube/graphene-combined enzyme interface are analyzed at the atomic level using molecular dynamics simulations. The morphology of the enzyme was visualized using an elastic network model by performing normal-mode analysis based on electrochemical and microscopic experiments. Single-carbon electrodes exhibited poorer electrical characteristics than those prepared as composites with enzymes. Furthermore, the composite interface exhibited 4.61- and 2.45-fold higher direct electron efficiencies than GOx synthesized with single-carbon nanotubes and graphene, respectively. Based on this study, we propose that pristine graphene has the potential to develop glucose oxidase interfaces and carbon-nanotube–graphene composites for easy fabrication, low cost, and efficient electrode structures for enzyme-based biofuel cells.

## 1. Introduction

Graphene and carbon nanotubes (CNTs) have found extensive applications in creating interfaces with enzymes and in the development of bioelectrodes as composite materials [1,2,3,4]. Although CNTs are commonly employed because of their cost-effectiveness, pristine graphene (excluding graphene oxide) has not yet been extensively utilized [5]. The noteworthy physical attributes of pristine graphene, such as its high electrical conductivity, mechanical properties, and transparency, have sparked interest in its potential as a bioelectrode. Tkac et al. reported the significance of employing graphene surfaces for advancing biofuel cell technology. However, they emphasized the necessity of exploring and understanding the full potential of graphene in this context [5].

The interfacial relationship between glucose oxidase (GOx) and the electrode surface is important in improving the performance of devices for glucose sensing and biofuel cell systems to generate energy from glucose [6,7,8]. In particular, electrodes made of carbon materials have been developed for electrode surface treatment using inexpensive manufacturing technology and synthesis with various materials [9,10]. Glucose oxidase oxidizes the glucose present in blood sugar to produce electrons. In particular, the flavin adenine dinucleotide (FAD) cofactor in GOx, which is defined as the active site, generates electrons by oxidizing glucose to hydrogen peroxide and d-glucose lactones. During this process, electron transfer occurs between the enzyme and the electrode, and the performance of the electrode is determined by the speed of electron transfer [6,7,11].

Electrodes composed of carbon materials can be broadly divided into single-walled carbon nanotubes (SWNT), multi-walled carbon nanotubes (MWNT), and graphene. Among these, CNTs have been used independently or synthesized with various materials because of their high productivity, low cost, and ability to aggregate well [9,10,12]. Graphene-based electrodes have been applied to the development of enzyme-based electrodes because of their high electrical conductivity; however, owing to their low productivity, they have been used less frequently than CNTs and have been developed in the form of graphene oxide, which is relatively easy to produce. CNTs are manufactured by chemical vapor deposition (CVD), which is much easier and has a longer manufacturing history than graphene. Graphene variants involve physical and chemical processes executed to produce graphene oxides and pristine graphene from graphite. In addition, an enzyme-based electrode was developed by synthesizing CNTs and graphene oxides to fabricate hybrid composites, which showed better electrode performance [1,13].

However, to the best of our knowledge, no studies have analyzed how CNTs and pristine graphene form biointerfaces and why they behave differently from GOx. To develop enhanced electron transfer performed electrode, interfacial analysis between enzyme and electrode material is necessary for the bottom-up development approach. This study analyzes how GOx interacts with CNTs and graphene variants at the molecular level and discusses whether the results are different when developed into a composite through electrochemical and microscopic experiments. By analyzing the interaction between enzymes and carbon materials, we obtained better results than those obtained by simply manufacturing a composite using only carbon materials and physically or chemically immobilizing the enzyme. Furthermore, in this study, we analyze the interactions between CNT and graphene and each electrode and enzyme and also synthesize two composite materials to analyze their interactions with enzymes, confirming that the properties of the existing electrodes can be improved.

To determine how GOx interacts with the CNTs and graphene at the molecular level, we performed molecular dynamics (MD) simulations and visualized the interfacial properties. MD has been used to analyze the behavior and shapes of biomolecules and has been applied to observe biomolecule–surface interfaces on various surfaces [14,15,16,17,18]. Modeling involved a comparison of carbon materials, specifically graphene oxide (GO), reduced graphene oxide (rGO), and carbon nanotubes (CNTs), to investigate structural surface alterations in the enzyme. Furthermore, a separate comparison within the CNTs was conducted to distinguish between the metallic and semiconductive properties. In addition, normal-mode analysis was performed by constructing an elastic network model for enzymes in bulk solution under immobilized conditions to define general protein vibrational motion. Finally, to verify the molecular-level simulation results, an enzyme-based electrode was fabricated, and its performance was confirmed by electrochemical experiments. Electrochemical experiments can measure the current density and electron transfer rate and have the advantage of being able to compare quantitatively the comparison group designed in this study and the results of other studies. This study shows the differences and advantages of CNT and graphene compared with other materials when synthesized with GOx and discusses future application directions. Furthermore, by appropriately synthesizing CNTs and graphene, it was verified that the CNT/graphene composite exhibits superior properties in enzyme-based electrodes, and it is expected to be used in the field of enzyme-based bioelectrodes by combining various enzymes and proteins.

## 2. Materials and Methods

### 2.1. Materials

The 1 M phosphate-buffered solution (PBS), KCl, K_3_Fe(CN)_6_, D+ glucose, GOx, FAD, Tween 20, and single-walled carbon nanotube (SWNT) powder were purchased from Sigma–Aldrich (St. Louis, MO, USA), and pristine graphene monolayer flake solutions were purchased from the Graphene Supermarket (Ronkonkoma, NY, USA).

### 2.2. Apparatus

All the electrochemical measurements, including cyclic voltammetry (CV), were performed at room temperature (20~25 °C) using a CompactStat (Ivium Technologies BV, North Brabant, The Netherlands). An S-4800 microscope (Hitachi, Tokyo, Japan) was used for the field-emission scanning electron microscopy (SEM) images. Transmission emission microscopy (TEM) images were obtained using a Tecnai 20 (FEI Corp., Hillsboro, OR, USA).

### 2.3. GOx/Carbon Nanocomposite Electrode Preparation

GOx (glucose oxidase) and FAD (1 mM) were mixed with distilled water. CNT (0.5 mg) and 20 µL of Tween 20 was mixed in 5 mL of distilled water. Bath sonication was conducted for 30 min to break down large aggregates. Ultrasonication was conducted at 20 kW for 1 h in an ice bath. The sonicated solution was treated using an ultracentrifuge (10,000× *g*, 4 °C, 20 min) to remove aggregation. The supernatant solution was extracted and stored at 4 °C in the refrigerator. Dispersed CCNT (0.1 mg/mL) and graphene solution (0.001 mg/mL) were mixed with the GOx-FAD solution. The mixture was sonicated using ultra-tip sonication at 20 kW for 1 h in an ice bath. The sonicated solution was stored at 4 °C in a refrigerator.

### 2.4. SEM/TEM Sample Preparation

The GOx/carbon nanocomposite solution was dropped onto a Si wafer and stored under 100% humidity conditions for 1 d. The Si wafer was then gently washed with pure water and dried using a nitrogen gun. The GOx/carbon nanocomposite solution was dropped onto a holey carbon TEM grid and dried with a nitrogen gun.

### 2.5. CV Characteristics Measurements

The GOx/carbon nanocomposite solution was dropped on the screen printed electrode (SPE) and stored in 100% humid conditions for 1 d. The electrodes were then gently washed with pure water. Prepared K_3_Fe(CN)_6_ (0.1 M), KCl (0.1 M), and PBS (0.1 M) solutions were dropped on the electrode. CV was measured under various conditions; potential range between −0.3 and 0.6 V, E onset at 0 V, E step of 10 mV, and scan rates of 10, 20, 35, 50, 65, 80, 100, and 120 mV/s. To verify the GOx-FAD mechanism, a glucose solution (0.1 mg/mL) was applied, and CV was performed under various conditions: −1 and 0 V, E onset of −1 V, E step of 10 mV, and scan rates of 10, 20, 35, 50, 65, 80, 100, and 120 mV/s.

### 2.6. Molecular Modeling: GOx, CNT, and Graphene

GOx is a dimeric protein composed of two identical polypeptide chain monomers that are noncovalently linked to each other and an active site where glucose binds is located inside. To simulate GOx at the atomic level, GOx from the *A. niger* (PDB ID: 1CF3) model was selected from the protein data bank (PDB). Missing hydrogen bonds were added using the software GROMACS 2019. The coenzyme FAD was submitted into PatchDockserver with binding residue indices of 29, 30, 50, 98, 103, 107, 110, 250, 549, and 561 for 1CF3 collected from the OPLS/AA force field, similar to our previous study [16].

To explore the mechanism of immobilization on CNT and graphene surfaces, CNTs with three different chiralities were constructed using the Visual Molecular Dynamics (VMD) 1.9.3 software and (6, 6), (6, 5), and (7, 6) chiral CNTs. CNTs display different electrical properties depending on their chirality [19]. For the development of GOx–carbon electrodes, it is essential to examine the binding properties associated with different chiralities of CNTs. Pristine graphene and graphene congeners, such as graphene oxide (GO) and reduced graphene oxide (rGO), were constructed with sizes of 6 × 6 nm^2^, which match the enzyme dimer size in our previous studies [14]. Graphene sheets with an armchair orientation were constructed using the VMD nanotube builder. The topology parameters for graphene were generated using an Automated Topology Builder, and hydrogen atoms were added to the sheet edges [20]. For GO and rGO, the Lerf–Klinowski model, which reflects the typical results of the oxidation process, was utilized [4,21]. The molecular compositions of GO and rGO were referenced from our previous publication [14].

### 2.7. MD Simulations

The GROMACS 2019 software was used to analyze the interface between the GOx and the carbons using the forcefield of OPLS/AA. The periodic simulation boxes were positioned 12 Å away from the enzyme–electrode interface and were modeled using cubic boxes. For the water model, simple point-charge water molecules were introduced into these cubic boxes. The salt concentration was maintained at 0.15 mol L^−1^, and counter-ions (Na^+^ and Cl^−^) were added to neutralize the system. The energy minimization process was carried out up to 10,000 iterations using the steepest descent method, with a tolerance of 1 kJ mol nm^−1^, to stabilize the interface and allow water molecules to move freely. Equilibration was achieved through 100 ps MD simulations using an isothermal–isochoric (NVT) ensemble, followed by another 100 ps using an isothermal–isobaric (NPT) ensemble at 1 bar and 300 K, respectively. Long-range electrostatic interactions were computed using the particle-mesh Ewald method, while short-range and van der Waals (vdW) interactions were applied with a cutoff distance of 12 Å. Finally, molecular dynamics simulations were conducted without any constraints after the stabilization process, with a 50 ns trajectory applied to the six different interfaces. All trajectories were recorded at 1 ps intervals to ensure reproducibility, with each of the six simulations performed three times.

### 2.8. Data Analysis for MD

To evaluate the structural properties of the enzyme, the root-mean-square deviation (RMSD), number of contact atoms between the enzyme and the carbon material surfaces, radius of gyration of the enzyme, and minimum distance between the enzyme and the surface were calculated. All data were analyzed using the GROMACS 2019 plug-in. For the binding analysis, the binding energy of the GOx–carbon material interface was calculated using molecular mechanics/Poisson–Boltzmann surface area (MM/PBSA) analysis for the last 1 ns of each run. Our previous studies used MM/PBSA for enzyme–ligand, protein–protein, and enzyme–surface interfaces [22,23,24]. This was calculated using the GROMACS plug-in “g_mmpbsa” [25]. A detailed description of the binding energy calculations is provided in the Appendix A.

### 2.9. Elastic Network Model and Normal-Mode Analysis of Enzyme

An elastic network model (ENM) was employed to visualize the vibrational mode shape of GOx based on detailed atomic-level structural information. This model was initially proposed by Tirion et al. and Hinsen et al. [26,27]. The springs maintain their initial lengths, denoted by the reference distance *r*_0_. The interaction between pairs of atoms is determined by a constant *k* or a function $k(r_{ij})$ decaying with distance $r_{ij}$, which is tunable by a constant $r_{0}$. We utilized the ENM formulation considering all atoms except hydrogen atoms. The spring constants between the atoms, which represent the interatomic interaction strengths, were weighted by their initial distances.
(1)$$k\left(r_{ij}\right)=k_{0}\exp\left[1-{\left(\frac{r_{ij}}{r_{0}}\right)}^{2}\right]$$

The formula for the spring constants is as follows: *k*_0_ = 100 kcal/(mol·Å^2^), $r_{ij}$, is the distance between atoms in a pair, and $r_{0}$ = 5 Å is a distance-weight parameter. Interactions between atoms were smoothly cutoff within 12 Å. Previous studies have demonstrated that the lower-frequency normal modes derived from the ENM align well with experimental measurements or standard empirical force-field simulations. These modes collectively entail significant amplitude motions and structural alterations in proteins, which play crucial roles in shaping their physiological functions and influence their mechanical and thermodynamic properties.

After setting up the ENM, the normal modes were calculated by diagonalizing the mass-weighted Hessian matrix, which represents the second derivative of the potential energy.
(2)$$H_{ij}=\frac{\partial^{2}E}{\partial x_{i}\partial x_{j}}$$
where *E* is the potential energy of the system, and $x_{i}$, $x_{j}$ are the fluctuations of the atomic positions on atoms *i* and *j*. The eigenvalues of the mass-weighted Hessian matrix $H^{\prime }=M^{-1/2}HM^{-1/2}$ provide the frequencies of corresponding modes, where M is a diagonal matrix of atomic masses. The corresponding eigenvectors of $H^{\prime }$ describe the shapes of the vibrational modes. To understand the construction of the ENM, a relevant figure is provided in Appendix A.

## 3. Results and Discussion

### 3.1. Atomics Scale Simulations for GOx-CNT/Graphene Interfaces

To compare differences when GOx was physically combined on the surface of SWNT and graphene, six simulation systems were configured, as shown in Figure 1 and Appendix A. Graphene, GO, and rGO were designed to have the same areas, and their distances from the surface at the beginning of the simulation were the same. Many studies have been conducted on graphene oxide, rather than on pristine graphene-based enzyme electrodes, as it contains various oxygen-containing functionalities such as epoxide, carboxyl, hydroxyl, and carbonyl [28,29,30]. GO is more hydrophilic than pristine graphene; in the past, we had proven that the laccase enzyme, which has many hydrophobic amino acids on its surface, has a stronger binding affinity for pristine graphene than for gold or CNT [14,15]. Therefore, we hypothesized that hydrophobic or electrostatic interactions affect the binding affinity of the GOx–carbon material interface differently. Furthermore, CNTs exhibit different properties owing to their chirality. Thus, we modelled metallic (6, 6), semiconducting (6, 5), and (7, 6) CNT to determine whether the chirality of CNT affects the immobilization of GOx.

As shown in Figure 1, GOx was stably immobilized on graphene and CNTs. The figure shows the last part of the MD trajectory with a total duration of 50 ns. Appendix A shows that the six MD systems become saturated after 7–50 ns. Appendix A shows that the enzyme and surfaces were immobilized throughout the MD trajectories, thus confirming the minimum distance between the enzyme and the surfaces. The RMSD values of the GOx-G and GOx-CNT(6, 6) systems fluctuated more than those of the others, but the RMSD values of the enzyme saturated well, as shown in Figure 2a and Appendix A. In a previous study by Ghoshdastider et al., the specific binding site of GOx on the graphene surface was determined by MD simulations [31]. For the initial state of all the MD simulations, we set up carbon surfaces immediately below the specific binding site. To analyze the structure and shape of the enzyme after binding to the surface of the carbon material, the RMSD and radius of gyration (Rg) were determined (Figure 2). RMSD values represent total conformational change during MD trajectories and Rg refers how much enzyme maintains globular morphology. Both values can explain the conformational change in enzyme during immobilization. In Figure 2a, the RMSD value was analyzed for the entire MD trajectory, and Figure 2b shows the average value for the last 1 ns after the stabilization. For each system, the RMSD values were saturated after a simulation time of 10 ns. However, the average protein’s RMSD values were different; GOx-G and GOx-GO yielded the highest values at ~0.33 nm and that of GOx-rGO was 0.24 nm. In our previous study on laccase and graphene congeners, we found that stronger hydrophobic interactions on the surface changed enzyme conformation and resulted in higher RMSD values. However, GOx showed that the hydrophobic interactions of graphene and the electrostatic interactions of GO affected the enzyme conformation at a similar level. Other researchers proposed that the lysine residue of GOx undergoes electrostatic interactions, thus resulting in greater conformational changes [32,33]. Therefore, the RMSD values of GOx-rGO and GOx-CNT(6, 5) were approximately 0.2 to 0.25 nm which represents that the reduced hydrophobic interaction of rGO and reduced binding surface of CNT affects similar amounts of morphological enzymatic changes. In previous studies, the laccase enzyme was immobilized on G, GO, and rGO surfaces by 0.019–0.039 nm [14]. Additionally, distances in the range of 0.40–0.43 nm were also observed in cases of enzymatic bindings to charged self-assembled monolayers [34]. RMSD values of GOx were 0.25 and 0.29 from our previous study on enzymes in bulk solution and binding to the graphene surface was obtained using different force fields [16]. The results showed that GOx exhibited a greater conformational change than laccase because of its larger size and dimeric shape. The MD results of this study seem to be reasonable considering the reference study outcomes.

The radius of gyration (Rg) in Figure 2c was extracted using the same method. A higher Rg value represents a lower conformational change owing to surface interactions, and lower Rg values correspond to a flattened morphology of the enzyme owing to greater surface binding affinity. The maximum GOx enzyme shape changes were observed in G and GO because of their high RMSD and low Rg values. This supports the finding that similar to previous studies, GOx attaches more strongly to surfaces with hydrophobic characteristics and changes the shape of the enzyme. GOx binds more strongly to the surfaces of graphene congeners than to CNTs. However, for graphene-CNT composites, the binding of the enzyme to graphene accelerates composite formation, as confirmed by the SEM images. The enzyme works as a mediator between graphene and CNT forest by hydrophobic and π–π interactions that enhance stability and the thickness of the composite material.

To assess properly the binding affinity and electron transfer rate of GOx to the surface, we calculated the number of contact atoms between the enzyme and the surface, minimum distance of the FAD-binding site to the surface, and the binding energy to the surface. As shown in Figure 1, the binding affinities of enzyme residues to two-dimensional graphene and its congeners are higher than that to CNT. Figure 3a shows that the contact atoms for GOx-G and GOx-rGO were 10,105 and 10,154, respectively. However, GOx-GO appears to have a lower amount of contact atoms (equal to 7956). This indicates that surface contact is mostly affected by hydrophobic interactions rather than electrostatic interactions. Figure 3c shows the binding energy analysis outcomes indicating that the vdW interaction is almost 3-fold higher than the electrostatic interaction, which affects the binding energy tendency. Furthermore, the distance between the binding site of the cofactor FAD and the surface (in all MD trajectories) yielded the closest values at the G and rGO surfaces. Thus far, the distance between the active site and the surface has been a critical value for determining the electron transfer rate according to the Marcus theory [14,35,36,37]. Therefore, the hydrophobic interaction at the graphene-GOx interface not only increases the binding affinity but also improves the electron transfer rate. It is worth noting that graphene and CNT composites may have a better potential for GOx immobilization than easily fabricated GO and oxide-functionalized CNT, based on our atomic-scale simulation results. Thus far, various studies have been conducted using GO and oxide-functionalized nanotubes; however, herein, we focus on the application of pristine graphene and nonfunctionalized CNTs.

In addition, CNTs were modeled with different chirality to determine whether metallic and semiconducting nanotubes affect immobilization differently, as shown in Appendix A. CNT(7, 6) has a larger diameter than the others, which lowers the Rg value of the enzyme. However, when comparing the three different chirality, the binding energies were not significantly different, as shown in Appendix A which are −378 to −400 kJ/mol.

### 3.2. Morphology Analysis of Enzyme Interface Using Elastic Network Model

In addition, we applied an elastic network model and normal-mode analysis to investigate the general conformational shapes and vibrational behavior of GOx depending on the bulk solution and immobilization conditions. Analyzing the vibrational behavior of proteins has been a great approach for the study of the general morphological features of Gox [38,39,40,41]. Additionally, we considered that GOx is different from laccase, which has a binding site for “glucose”. Laccase accepts electrons from the T1 copper site, and the electrons move through the pathway to reach the T2/T3 site [42,43]. However, it has been established that the glucose-binding site is located inside each monomeric chain [6,44]. Thus, we hypothesized that morphological changes near binding pocket of GOx would appear when compared with other enzymes or proteins. We calculated the natural frequencies of the first three modal shapes by solving the ENM eigenvalue problems, as shown in Figure 4. The corresponding modal shapes are represented in Figure 5. Figure 5a,d shows the first mode, Figure 5b,e shows the second mode, and Figure 5c,f shows the third mode. Figure 5a–c vs. Figure 5d–f show the opposite directions for the three modes. We found that the first three mode shapes did not change, regardless of whether they were in the bulk solution or the immobilization phase. The detailed vibrational shapes can be verified from the movies in the Appendix A.

Figure 4 shows that the natural frequency of GOx in the bulk solution is lower than that in the immobilized condition, except for the first mode of GOx-CNT(6, 5). The natural frequency is calculated through the following simple formula.
(3)$$\omega_{0}=\sqrt{\frac{k}{m}}$$
where $\omega_{0}$ is the natural angular frequency, k is stiffness, and m is mass. When the natural frequency increases, these replies increment of general stiffness or decrement of general mass of system. Interestingly, the natural frequency increased after binding onto the carbon surfaces, as shown in Figure 4. It can be noted that immobilization makes the enzyme stiffer and more stable in terms of its morphological properties. To be specific, general stiffness increment refers fluctuation of residues on enzyme and probability of movement during immobilization decline. In addition, for the first three modes, the values of the natural frequencies were higher for graphene and its congeners than for CNT. The tendencies of the ENM-NMA results were similar to the RMSD values of the MD simulations, where the values for GOx-G and GOx-GO were higher at the 1st and 3rd modes than the other modes. With respect to the modal shape (Figure 5), chains A and B rotate in the opposite direction of the *X*-axis for the 1st mode. For the 2nd mode, chains A and B rotate as the *Y*-axis pivots, and one of the chains opens, while the other closes. The 3rd mode is interesting in that it opens the glucose-binding site and works like a clam shell, opening its mouth. To properly bind glucose to each chain for dimer GOx from *Aspergillus niger* (PDB: 1CF3), we believe that the 3rd mode should appear more often compared with the 1st and 2nd modes, which potentially affects the glucose binding activity to enhance electron transfer. Furthermore, when we simulated glucose-binding MD from our previous study, we found that only one of two glucose molecules bound to the pocket in two-thirds of the case simulations. Interestingly, the general modal shapes of GOx suggest substrate-binding characteristics considering its morphology; accordingly, we may potentially develop enzyme shapes to improve its activity. Therefore, enhancing the occurrence of the first and third binding modes by controlling the glucose-binding pocket, along with using a graphene sheet with high binding affinity, will enhance the performance of GOx-based carbon electrodes.

### 3.3. Confirmation of GOx-CNT/Graphene Structure

As mentioned previously, we investigated the binding affinity of GOx to CNT and graphene using MD simulations. GOx bound tightly to CNT, and the binding of GOx to graphene was more stable than that to CNT. Based on the results of the MD simulations, we fabricated GOx and CNT/graphene composites. Figure 6 shows top- and side-view SEM images of GOx-CNTs (Figure 6a,b), GOx-graphene (Figure 6c,d), and GOx-CNT/graphene (Figure 6e,f). The CNTs are shown as randomly stretched strands in Figure 6b and GOx was filled in. Comparing Figure 6b,d, the GOx-CNT layer is thicker than the GOx-graphene layer.

GOx-coated graphene is shown in Figure 6d. Because graphene is a two-dimensional (2D) material, the surface van der Waals forces flatten the graphene sheet. Owing to the electrostatic repulsion force, the CNT stood and GOx helped the CNT stand well by filling in the spaces. The amount of GOx in the CNT structure appears to be much greater than the amount of GOx in the graphene structure. To immobilize GOx at the maximum quantity and density on the electrode, not only is the binding affinity to carbon materials (monomer to monomer at the atomic scale) important, but also the material coating mechanism based on the characteristics of carbon materials. Therefore, to fabricate the GOx-CNT/graphene composite in Figure 6e,f, we expected three-dimensional (3D) structural support by CNT and a dense coating of GOx by graphene. Thus, the enzyme acts as a mediator in the production of carbon composites. Chen et al. developed an enzyme-based biofuel cell using carboxylated CNT adsorbed onto 3D graphene [45]. This resulted in covalent immobilization of the enzyme onto the surface and a high surface area. Amphiphilic carboxylated CNT strongly interacted with 3D graphene based on hydrophobic and π–π interactions. Palanisamy et al. also noted that MWNT and GO hybrid composites prepared based on π–π and electrostatic interactions were critical for the absorption of GOx and hybrid composites [28]. From our microscopic experiments and the MD results listed above, it is not just the high surface area of carbon nanotubes (single- or multi-walled) that enhances the current density of the fabricated electrode, but also the binding of the enzyme to graphene or graphene oxide that is more important at the atomic scale. DET results from the binding of the enzyme to the two-dimensional graphene sheets, and CNTs act as pillars for the formation of carbon composites. Thus, the enzyme may be adsorbed on the graphene surface and CNT, which act as pillars in the hybrid composites.

As shown in Figure 6f, the GOx layer was the thickest. Appendix A shows a comparison of the SEM images without GOx. Without GOx, a layer of the structure is barely formed. The TEM image of CNT/graphene in Appendix A shows that the CNT and graphene can also bind to each other. GOx has a role in electron transfer (by digesting glucose) and also supports the coating structure. GOx corresponds to cement, graphene corresponds to the floor, and CNT corresponds to the columns in the building structure. Therefore, by mixing GOx, CNT, and graphene, we created a dense and thick GOx layer.

### 3.4. Evaluation of the GOx-CNT/Graphene Composite Electrode

To evaluate the GOx-CNT/graphene composite as a cathode, we investigated the current density and electron transfer rate using cyclic CV. Using the Fe(CN)_6_ reduction test in Appendix A and Figure 7a, we confirmed the current density of the carbon materials. As more carbon material is added, a higher current density is expected. CNT and graphene commonly contain a hexagonal carbon ring that acts as an electron–hole pairs. A large number of electron–hole pairs provide electron reduction reactivity for redox reactions. As shown in Figure 7a, the current density of the CNT/graphene-GOx composite was higher than that of the other samples. According to the SEM image in Figure 6, the graphene-CNT-GOx composite formed the largest number of electrons and holes. Compared with Appendix A without GOx, the structural stability of the carbon material was not guaranteed, so a low current density of Fe(CN)_6_^−4^ was recorded. Therefore, a thick layer of carbon material, GOx, is an important supporting mediator of the coating structure.

Subsequently, we tested the glucose-digestion ability of GOx, as shown in Figure 7b. As expected, the CNT/graphene-GOx composite exhibited the highest current density. Moreover, compared with CNT-GOx or graphene-GOx, the CNT/graphene-GOx current density increased. As shown in Figure 7a, the current density is slightly enhanced. This phenomenon led to a synergistic structural support effect between GOx and carbon materials. As mentioned previously, CNT, graphene, and GOx formed co-supporting materials which leads to a much higher glucose reduction reaction for GOx. Therefore, mixing various dimensional materials with GOx may help form a dense and thick coating layer on the electrode.

To validate the electrochemical performance of the CNT/graphene-GOx composite, comparisons were made with commonly reported catalysts, including bilirubin oxidase-based electrodes and metal oxide catalysts. Bilirubin oxidases are frequently used as substitutes for GOx in enzyme-based anode applications. Several studies have reported that bilirubin oxidase-based electrodes exhibit a current density of 300 to 600 μA/cm^2^ [46,47]. This indicates that the performance of the CNT/graphene-GOx composite is significant, considering its current density and electron transfer rate. In contrast, metal oxide catalysts, such as those used in universal chemical-induced tensile strain tuning-based oxygen-evolving electrocatalysis on perovskite oxides, show a current density range of 50–200 mA/cm^2^ [48,49]. This is on a different scale compared to our enzyme-based electrode, which exhibits a current density of approximately up to 1 mA/cm^2^.

As shown in Figure 8, we tested the GOx composite electrode by gradually increasing the scan rate and calculated the electron transfer rate using the Marcus theory of electron transfer [35,50]. The inset graph in Figure 8 provides variations in the peak current potentials and current densities with respect to the scan rate. The detailed methodology is provided in the Appendix A. Quantitatively, the electron transfer rate was also the highest for the CNT/graphene-GOx electrode, which was 4.61- and 2.45-fold higher than those of the CNT-GOx and graphene-GOx electrodes, respectively. The electron transfer rate indicates that the binding affinity results from MD simulations show that GOx-CNT is inferior to graphene. As shown in Figure 3c, the binding energy of GOx-G is 3.5 times higher than that of GOx-CNT(6, 5), and the electron transfer rate differs by 1.88 times. Although the values of binding affinity at the atomic scale and electron transfer rate obtained from experiments differ, both MD simulations and experiments demonstrate the significant role of graphene. Therefore, we confirmed that the electron transfer efficiency of the CNT/graphene-GOx electrode was the best and proved that it had the densest and thickest coating structures.

## 4. Conclusions

In conclusion, this study presents a comprehensive investigation into the atomic-scale simulations, structural confirmation, and evaluation of CNT/graphene-GOx interfaces, focusing on the immobilization of glucose oxidase (GOx) on various carbon materials. This study involved six simulation systems, each revealing valuable insights into the interactions between GOx and different carbon substrates. Atomic-scale simulations have shown that GOx exhibits distinct binding characteristics with a higher affinity for hydrophobic surfaces and the ability to alter its shape upon binding. Notably, GOx displayed stronger interactions with pristine graphene by hydrophobic and π–π interactions, which was confirmed based on the analyses of RMSD, Rg values, and binding energy. Furthermore, by performing ENM-NMA to determine the general vibrational modal shape, we found that the binding characteristics of glucose appeared to be more familiar to the third mode, and general stiffness, which is correlated to the stability of enzymatic function, increased during immobilization to the graphene surface. In addition, CNTs were found to work as pillars in CNT/graphene composites, and when an enzyme was applied for the fabrication of the hybrid material, a thicker coating of the composite was possible on the surface, and enhanced electrochemical properties were achieved.

Confirmation of the CNT/graphene-GOx structure demonstrated the stable immobilization of GOx on both CNTs and graphene, with notable differences in the structure and layer thickness between the two. The combination of GOx with CNTs and graphene created a 3D structural support, allowing the formation of a dense and thick GOx coating. This structural support was found to be crucial for achieving optimal electrode performance. Evaluation of the GOx-CNT/graphene composite electrode using various tests highlighted its excellent electron transfer rates and glucose-digestion capabilities. The composite electrode outperformed the other configurations, exhibiting a superior electron transfer efficiency and a denser coating structure. This study underscores the significance of combining various dimensional materials, such as GOx, CNTs, and graphene, to create effective enzyme-based electrodes.

In summary, the findings of this study provide valuable insights into the binding characteristics and performance of GOx with various carbon materials. These insights can inform the development of more efficient and effective enzyme-based electrodes, thereby contributing to advancements in biosensing and electrochemical applications.

Biosensors for Glucose Monitoring: The insights gained from this study can be directly applied to the development of high-performance biosensors for glucose monitoring, which are crucial for diabetes management. The enhanced electron transfer rates and stability of the GOx-graphene-CNT composites make them ideal candidates for continuous glucose monitoring systems (CGMS). These systems require sensors that can provide accurate, real-time glucose measurements with high sensitivity and stability over extended periods. The superior electron transfer efficiency and structural stability of the GOx-graphene-CNT composite electrodes can lead to more reliable and longer-lasting sensors.Biofuel Cells: Enzyme-based biofuel cells (EBFCs) convert biochemical energy from glucose into electrical energy. The high electron transfer rates observed in the GOx-graphene-CNT composites can significantly improve the power output of these biofuel cells. The stability and efficient electron transfer of the immobilized GOx ensure sustained catalytic activity, which is essential for maintaining the performance of EBFCs over time. This can lead to the development of more efficient and cost-effective biofuel cells for powering small electronic devices and potential applications in medical implants and portable power sources.Environmental Monitoring: Beyond healthcare, the GOx-graphene-CNT composites can be utilized in environmental monitoring to detect glucose and other analytes in water and soil samples. The high sensitivity and specificity of the enzyme-based sensors can help monitor pollution levels and detect contaminants in the environment, contributing to better environmental management and protection.

The comprehensive approach used in this study, from simulation to structural confirmation and performance evaluation, offers a holistic understanding of the CNT/graphene-GOx interface and its potential for various applications. Furthermore, we plan to apply atomic scale visualization technique and experimental confirmation to other materials not only to enzyme–carbon materials but also to various protein–metallic or semiconducting materials. We believe that our confirmation of enzyme–carbon interface analysis at the atomic scale level and electrochemical approach will shed light on fabrication of GOx-based bioelectrode field.

## Figures and Tables

**Figure 1 nanomaterials-14-01073-f001:**
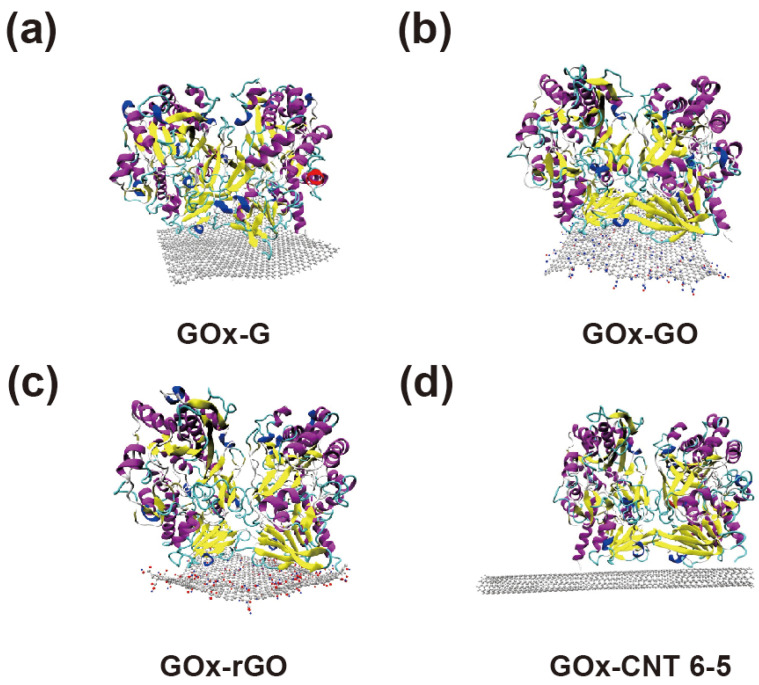
Immobilization of glucose oxidase (GOx) on G, graphene oxide (GO), reduced GO (rGO), (**a**–**c**) and carbon nanotube (CNT) (**d**). The final trajectories of the 50 ns molecular dynamics (MD) simulations are shown.

**Figure 2 nanomaterials-14-01073-f002:**
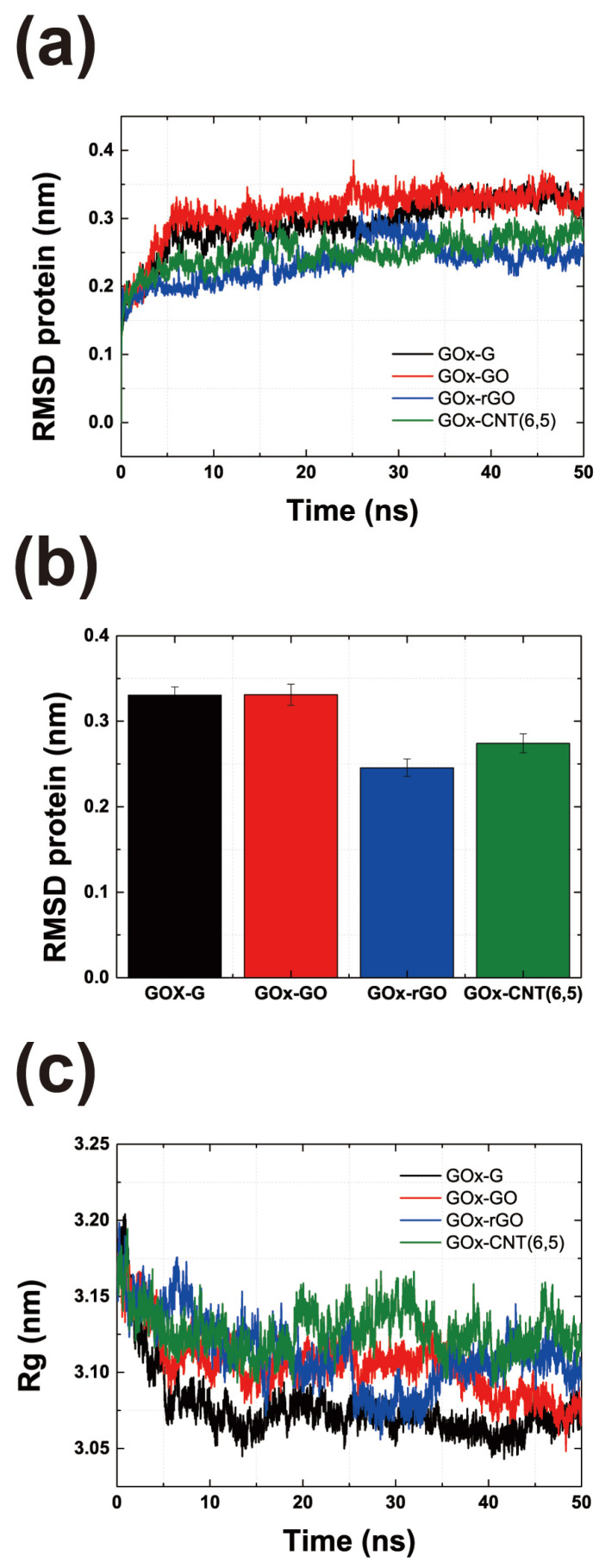
Conformational analysis of GOx on carbon material surfaces. (**a**) Root-mean-square deviation (RMSD) values of enzymes obtained through entire MD trajectories. (**b**) RMSDs of enzymes during the last 1 ns of the 50 ns MD trajectories. (**c**) Radius of gyration of enzymes during the last 1 ns of the 50 ns MD trajectories.

**Figure 3 nanomaterials-14-01073-f003:**
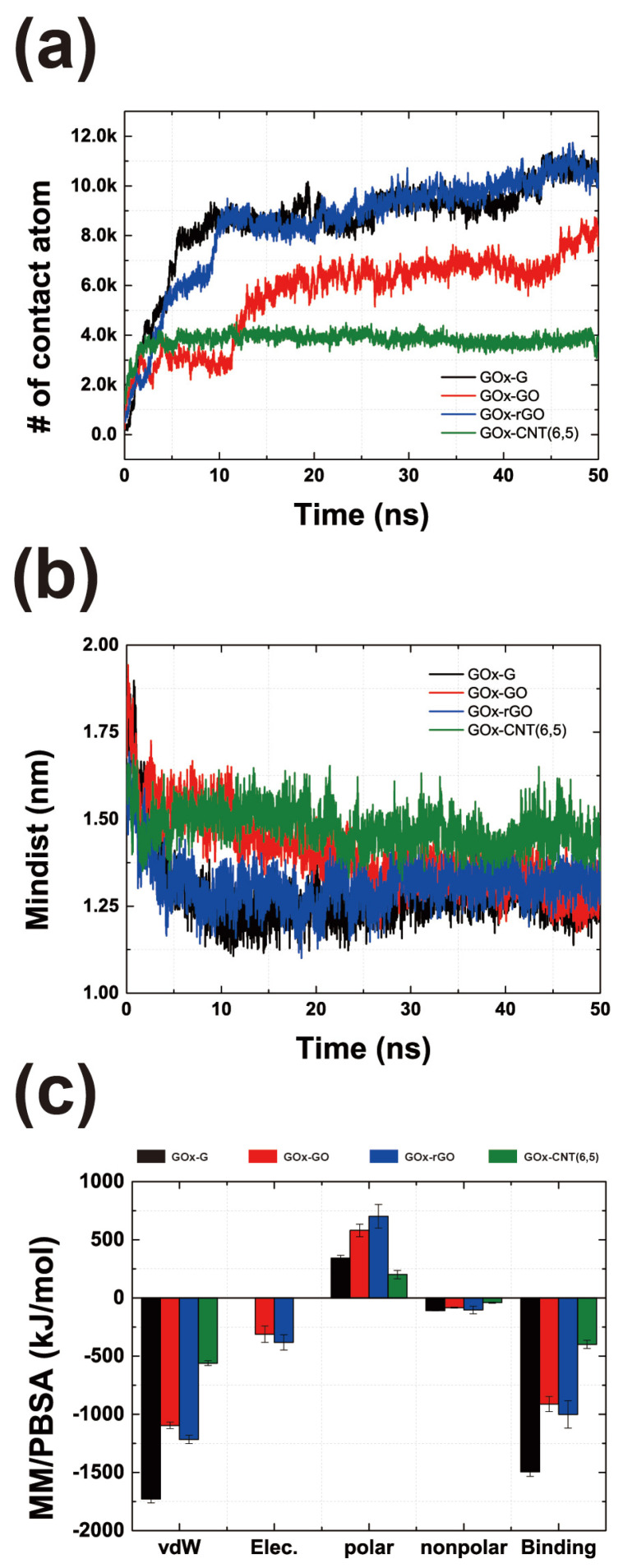
Binding characteristics of GOx on carbon material surfaces. (**a**) Time-dependent representation for several contact atoms of enzyme surface interfaces. (**b**) Average value of the minimum distance of the final 1 ns of the MD trajectories. (**c**) Molecular mechanics/Poisson–Boltzmann surface area (MM/PBSA) binding calculation. Average outcomes obtained during the final 1 ns MD trajectories are shown.

**Figure 4 nanomaterials-14-01073-f004:**
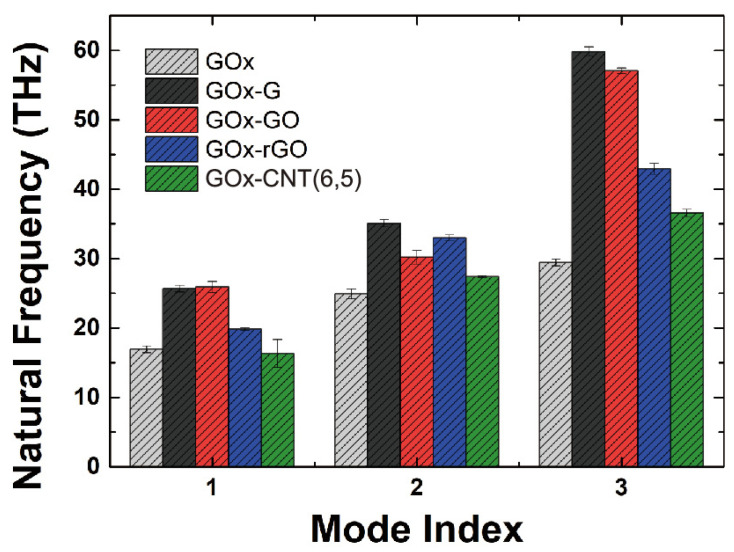
First three natural frequencies of GOx at different conditions from ENM-NMA.

**Figure 5 nanomaterials-14-01073-f005:**
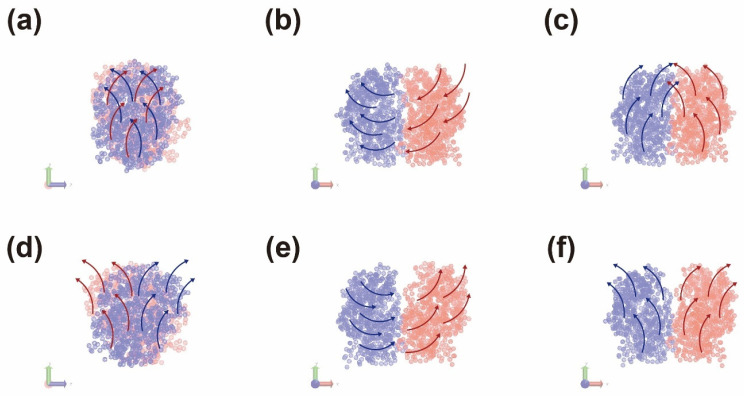
Normal modal analysis and shapes of GOx. (**a**,**d**) Modal vector of 1st mode. (**b**,**e**) modal vector of 2nd mode. (**c**,**f**) Modal vector of 3rd mode. Chain A is colored in blue and chain B is colored in red. Arrows are drawn for modal vectors.

**Figure 6 nanomaterials-14-01073-f006:**
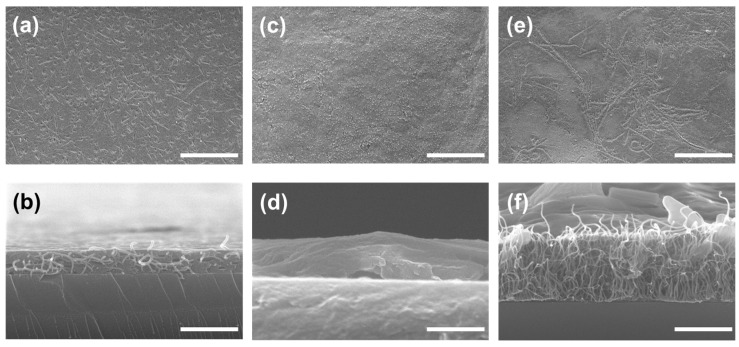
Top and side-scanning electron microscopy (SEM) views of CNTs + GOx (**a**,**b**), graphene + GOx (**c**,**d**), and graphene + CNT + GOx composite (**e**,**f**). The scale bars are 1 µm.

**Figure 7 nanomaterials-14-01073-f007:**
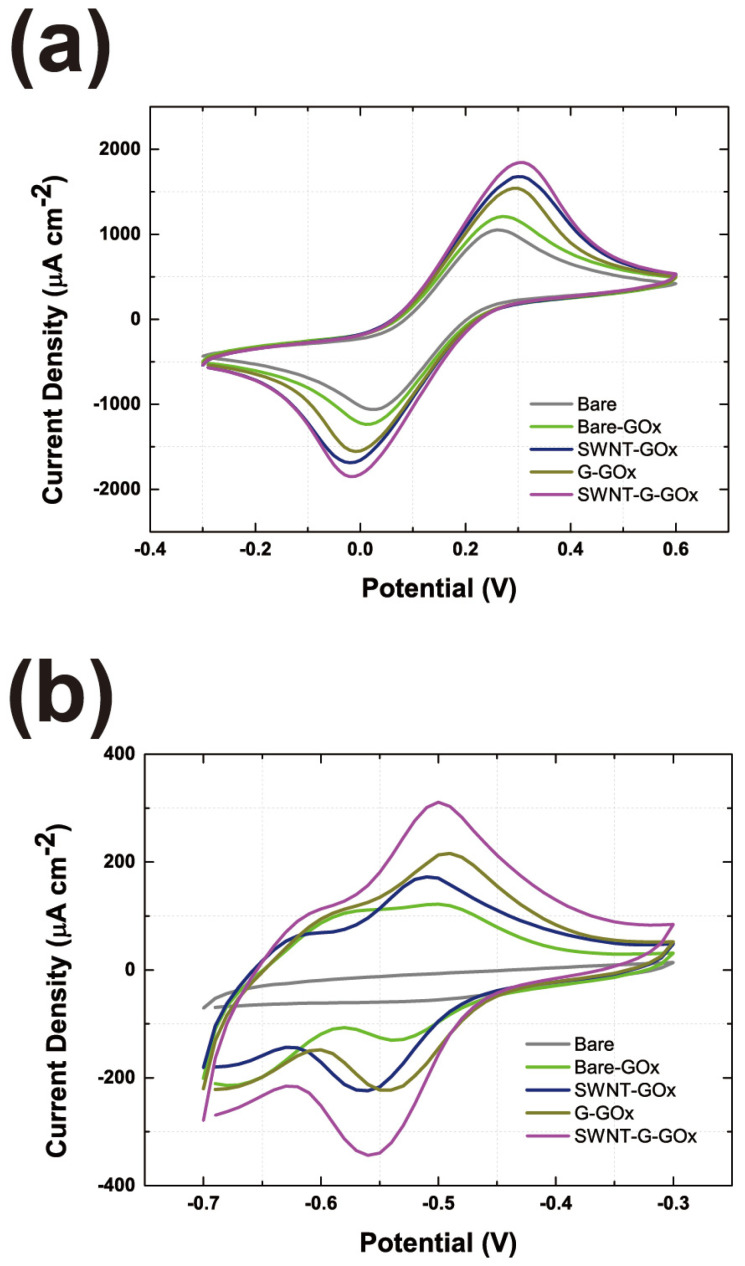
CV curves of Fe(CN)_6_^−4^ reduction experiments (**a**) and glucose-digestion experiments (**b**) according to enzyme and modified carbon materials (scan rate: 50 mV/s).

**Figure 8 nanomaterials-14-01073-f008:**
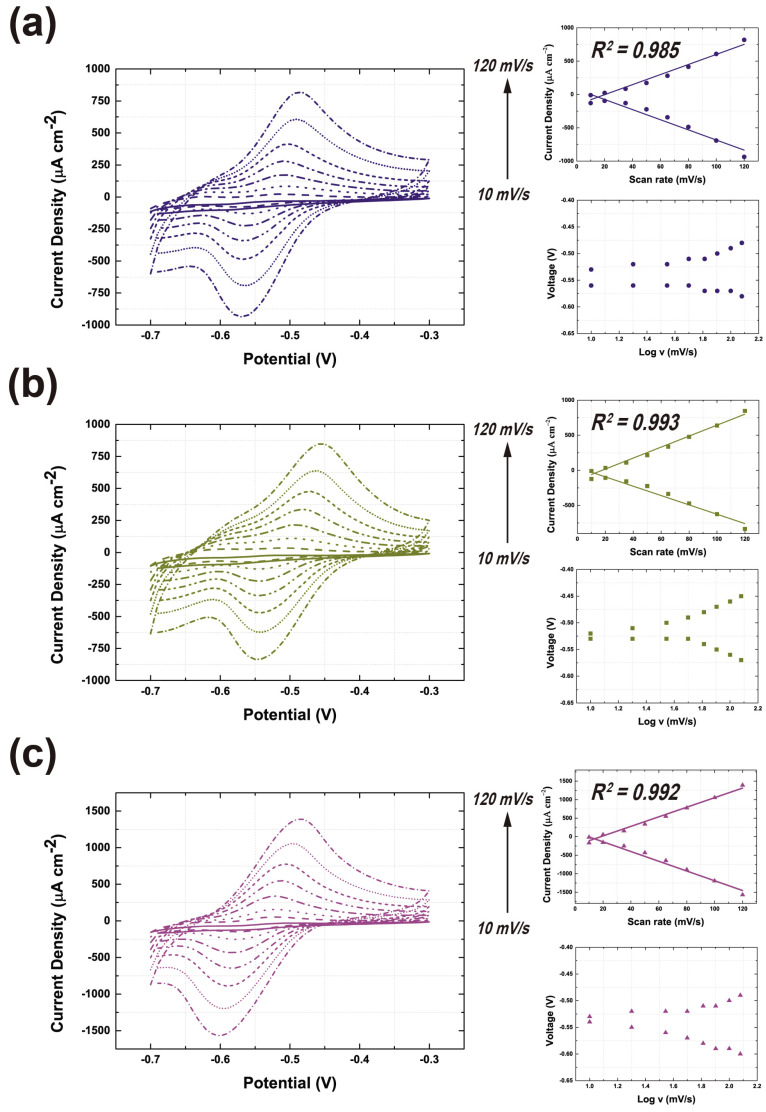
CV curves of the CNT-GOx (**a**), graphene-GOx (**b**), and CNT/graphene-GOx (**c**) at different scan rates (10, 20, 35, 50, 65, 80, 100, and 120 mV/s).

## Data Availability

Dataset available on request from the authors.

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
