# Peer review of "Investigation of Direct Electron Transfer of Glucose Oxidase on a Graphene-CNT Composite Surface: A Molecular Dynamics Study Based on Electrochemical Experiments"

_nanomaterials, 2024, doi:10.3390/nano14131073_

Round 1

Reviewer 1 Report

Comments and Suggestions for Authors

In this study, the authors presented the simulations to evaluate the interactions of glucose oxidase over the GOx/SWCNT composite and the related experimental results. First, a molecular model was developed to evaluate the affinity between glucose oxidase and carbon nanomaterials. From the molecular dynamic simulations, the affinity of glucose oxidase is better over GOx. Therefore, the authors designed electrochemical experiments to validate the model findings and the results were in good agreements with the simulations. Overall, this study presented a good theoretical procedure for experimental design and provided good scientific insights. Therefore, it is recommended for publication. However, some issues need to be addressed before publication and are listed below:

1.      In section 3.2, it is not quite clear how the authors draw the conclusion for the affinity between the oxidase and carbon structures. Please provide a more specific index for the readers. Was the RSMD in third mode?

2.      In the discussion of Figure 7 or 8, the peak potential differences for each material should be evaluated to address the number of electrons transferred. It seems those with CNTs are worse than the others. That can be a proof for the simulation of less affinity.

3.      Similarly, the authors inferred the electron transfer rates from Figure 8, and mentioned that in 432-435. However, there lacks the related equations or calculation method in the text or figures. Please revise the text and provide the slopes in the sub-figures.

Comments on the Quality of English Language

In this study, the authors presented the simulations to evaluate the interactions of glucose oxidase over the GOx/SWCNT composite and the related experimental results. First, a molecular model was developed to evaluate the affinity between glucose oxidase and carbon nanomaterials. From the molecular dynamic simulations, the affinity of glucose oxidase is better over GOx. Therefore, the authors designed electrochemical experiments to validate the model findings and the results were in good agreements with the simulations. Overall, this study presented a good theoretical procedure for experimental design and provided good scientific insights. Therefore, it is recommended for publication. However, some issues need to be addressed before publication and are listed below:

1.      In section 3.2, it is not quite clear how the authors draw the conclusion for the affinity between the oxidase and carbon structures. Please provide a more specific index for the readers. Was the RSMD in third mode?

2.      In the discussion of Figure 7 or 8, the peak potential differences for each material should be evaluated to address the number of electrons transferred. It seems those with CNTs are worse than the others. That can be a proof for the simulation of less affinity.

3.      Similarly, the authors inferred the electron transfer rates from Figure 8, and mentioned that in 432-435. However, there lacks the related equations or calculation method in the text or figures. Please revise the text and provide the slopes in the sub-figures.

Reviewer 2 Report

Comments and Suggestions for Authors

The author has conducted a comprehensive study on the interaction of glucose oxidase (GOx) with graphene and carbon nanotubes (CNTs), focusing on the direct electron transfer mechanisms at the molecular level using molecular dynamics (MD) simulations. The study includes detailed analysis of the structural, electrochemical, and morphological properties of GOx when combined with different carbon materials, as well as the fabrication and evaluation of enzyme-based electrodes. While the paper presents significant findings, there are several areas that require improvement and clarification. However, there are some aspects where revisions are recommended:

1.     While the MD simulations are thoroughly described, there should be a clearer justification for the choice of simulation parameters, such as the size of the graphene sheets and CNT chiralities. Why were these specific sizes and chiralities chosen?

2.     The manuscript should include a comparison of the electrochemical performance of the GOx-graphene-CNT composite with other commonly reported catalysts, such as metal oxide catalysts [Appl. Phys. Rev. 9, 011422 (2022); Adv. Mater. 2023, 2305074]. Relevant literature could be cited to provide context and highlight the relative performance advantages or disadvantages of the composite material compared to traditional catalysts.

3.     The discussion around the binding energy calculations could be expanded. What are the implications of the differences in binding energies observed? How do these energies correlate with the electron transfer efficiencies?

4.     The normal mode analysis (NMA) section is well-detailed, but the relevance of the NMA findings to the overall study could be more explicitly linked. How do the vibrational modes influence the electron transfer process?

5.     The potential practical applications of the findings should be discussed in more detail. How can these insights into GOx immobilization on graphene-CNT composites be translated into real-world bioelectrode applications?

Comments on the Quality of English Language

Minor editing of English language required
